# Adapting the Intensity Gradient for Use with Count-Based Accelerometry Data in Children and Adolescents

**DOI:** 10.3390/s24103019

**Published:** 2024-05-10

**Authors:** Christina J. Alexander, Sarah L. Manske, W. Brent Edwards, Leigh Gabel

**Affiliations:** 1Biomedical Engineering Graduate Program, Schulich School of Engineering, University of Calgary, Calgary, AB T2N 1N4, Canada; 2Human Performance Laboratory, Faculty of Kinesiology, University of Calgary, Calgary, AB T2N 1N4, Canada; 3McCaig Institute for Bone and Joint Health, Cumming School of Medicine, University of Calgary, Calgary, AB T2N 1N4, Canada; 4Alberta Children’s Hospital Research Institute, Cumming School of Medicine, University of Calgary, Calgary, AB T2N 1N4, Canada; 5Department of Radiology, Cumming School of Medicine, University of Calgary, Calgary, AB T2N 1N4, Canada

**Keywords:** accelerometer, cutpoint-free metrics, physical activity, ActiGraph, activity monitor, acceleration

## Abstract

The intensity gradient is a new cutpoint-free metric that was developed to quantify physical activity (PA) measured using accelerometers. This metric was developed for use with the ENMO (Euclidean norm minus one) metric, derived from raw acceleration data, and has not been validated for use with count-based accelerometer data. In this study, we determined whether the intensity gradient could be reproduced using count-based accelerometer data. Twenty participants (aged 7–22 years) wore a GT1M, an ActiGraph (count-based), and a GT9X, ActiGraph (raw accelerations) accelerometer during both in-lab and at-home protocols. We found strong agreement between GT1M and GT9X counts during the combined in-lab activities (mean bias = 2 counts) and between minutes per day with different intensities of activity (e.g., sedentary, light, moderate, and vigorous) classified using cutpoints (mean bias < 5 min/d at all intensities). We generated bin sizes that could be used to generate IGs from the count data (mean bias = −0.15; 95% LOA [−0.65, 0.34]) compared with the original IG. Therefore, the intensity gradient could be used to analyze count data. The count-based intensity gradient metric will be valuable for re-analyzing historical datasets collected using older accelerometer models, such as the GT1M.

## 1. Introduction

Accelerometers are commonly used to measure physical activity (PA). These devices are often worn on the wrist or hip, and track and store accelerations in real time, allowing researchers to examine PA over the time period it was worn [1]. Older accelerometer models with limited storage did not output raw acceleration data but rather a proprietary “count” value for each epoch (a user-selected length of time). Accelerometer calibration studies often used energy expenditure (indirect calorimetry to derive metabolic equivalent tasks (METs)) to develop count thresholds (cutpoints) corresponding to the intensity of PA [2]. Traditionally, these cutpoints were then applied to count data to identify how much time participants spent at different PA intensities (i.e., sedentary (SED), light PA (LPA), moderate to vigorous PA (MVPA), and vigorous PA (VPA)) [2,3]. A limitation of the cutpoint-based approach is that very rich datasets of PA (e.g., those sampled many times per minute) are reduced to a few measures of time spent at different intensities of PA (e.g., minutes per day of MVPA). Although this is a convenient way to convey daily PA recommendations to the public, it may limit our ability to detect relationships between PA and health outcomes. 

Recently, there has been increased interest in accounting for the full PA profile. For example, the intensity gradient (IG) describes the distribution of the full spectrum of PA intensity [4]. The IG was originally developed using the Euclidean norm minus one (ENMO) acceleration metric from wrist-worn accelerometers worn 24 h per day with a 5 s epoch. ENMO is a summary metric calculated using the vector magnitude of triaxial raw acceleration data (mg) that is averaged over user-defined epochs [5]. The IG involves sorting the participant’s full spectrum of PA data into bins on the basis of the intensity and using the log–log slope of the intensity of the PA and time accumulated at each intensity to quantify the distribution of the individual’s PA intensity (Figure 1). Slopes are always negative to reflect less time spent at higher intensities, and a steeper slope indicates an uneven distribution of time spent across all intensities (e.g., more time at lower intensities); a shallower slope indicates a more even distribution of time spent at all intensities of PA (e.g., more time at higher intensities). The IG metric has been used to examine the relationship between PA and bone mineral content (BMC,) areal bone mineral density (aBMD), and body mass index in children and adolescents [6,7,8]. However, most studies that have used the IG to identify associations between PA and health outcomes have used raw acceleration data.

Because large historical PA datasets have been collected using older accelerometer models that output data in counts, it is necessary to understand whether novel metrics developed using summary measures of raw acceleration data (such as the IG developed with ENMO) can be used with accelerometry count data (i.e., whether the IGs calculated using ENMO can be reproduced using count data). Because historical count data were collected on uniaxial ActiGraph devices that cannot store raw accelerations, such as the GT1M, it is impossible to compare IGs generated from ENMO and counts with these models. Accelerometer models that collect both counts and raw accelerations (i.e., the GT9X) can be used to compare IGs from ENMO and counts. Therefore, to ensure that IGs are applicable to historical count data, counts from the GT1M and GT9X must first be compared. Comparisons among different models of the ActiGraph accelerometer have found good agreement between different models in the past [9,10,11,12], but not specifically between the GT1M and GT9X. Therefore, the purpose of this study was to first determine whether the activity counts collected by an older accelerometer, GT1M (ActiGraph, Pensacola, FL, USA), were comparable with the activity counts collected by a newer accelerometer, the GT9X (ActiGraph, Pensacola, FL, USA). The second objective was to investigate whether the IG calculated from ENMO acceleration data could be replicated using activity count data. We hypothesized that the count data would be comparable between the two accelerometer models and that we would be able to reproduce the IG metric developed for raw acceleration data with count data. 

## 2. Materials and Methods

### 2.1. Study Design

The 20 participants for this cross-sectional validation study included healthy children, adolescents, and adults from Calgary, Alberta, Canada. We recruited participants by posting recruitment flyers at the University of Calgary, through the University of Calgary research portal, and by word of mouth. We obtained written informed consent from participants aged 14 years and older and written informed consent from the parents or legal guardians in addition to written assent for participants aged 13 years and younger. The University of Calgary Conjoint Health Research Ethics Board approved the study (REB22-0603).

### 2.2. Anthropometry

We assessed height using a stadiometer (model 213; Seca, Hamburg, Germany), with the participants standing barefoot on the scale, recording height measurements to the nearest millimeter. We measured body mass to the nearest 0.1 kg using a digital scale (model 874; Seca, Hamburg, Germany). All measurements were taken twice unless differences of >0.4 cm or 0.2 kg were detected, in which case, a third measurement was taken. The mean of 2 measurements or median of 3 was used in all analyses. 

### 2.3. Accelerometers

This study examined two ActiGraph accelerometers, the GT1M (ActiGraph, Pensacola, FL, USA) and the GT9X Link (ActiGraph, Pensacola, FL, USA). The GT1M is a small (3.8 × 3.7 × 1.8 cm), lightweight (27 g), uniaxial accelerometer that detects vertical accelerations of 0.05–2.50 g [13]. After digitization, this signal is passed through a band-pass filter (0.25–2.5 Hz) [13]. The signal is then used to output an activity count for each user-specified epoch that is representative of the acceleration during that time period. The GT9X is a smaller (3.5 × 3.5 × 1 cm), lighter (14 g) triaxial accelerometer that detects accelerations of ±8 g at 30–100 Hz [14]. The GT9X stores and outputs raw acceleration values. These raw accelerations can be converted into the epochs’ activity count data in ActiGraph’s proprietary software, ActiLife version v6.13.4 (ActiGraph, Pensacola, FL, USA). Both accelerometers can be worn on the wrist or hip.

For this study, we initialized the GT1M accelerometers to collect data in 15 s epochs, and used the GT9X accelerometers to collect data at 30 Hz in ActiLife v6.13.4 (ActiGraph, Pensacola, FL, USA). All accelerometers were initialized on the same computer. As per the manufacturers’ instruction for waist-worn accelerometers, we did not calibrate the accelerometers [14]. We attached the GT1M and GT9X onto an elastic belt side by side, and taped the GT9X and its clip to the belt to prevent it from slipping along the belt. Participants were instructed to wear the belt over light clothing or under their clothing, with the accelerometers positioned over their right hip.

### 2.4. Data Collection

The participants attended a 1-h in-lab testing session, during which anthropometry was collected and the following activities were completed: walking, brisk walking, jogging, sprinting, vertical jumping, jumping jacks, side-to-side shuffling through an obstacle course, climbing stairs, and cleaning up (picking up and putting back down pylons); all pacing was self-selected. These activities were chosen so that the participants would engage in a wide range of activities during the session, including different intensities of activities and various movements (e.g., vertical and side-to-side). 

After the session, the participants were sent home with a second set of accelerometers (GT1M and GT9X) and a PA log sheet. The participants were instructed to wear the accelerometers for three days during all waking hours and to take them off for any water activities. The participants were also asked to record when they put the accelerometer on in the morning and took it off at night, as well as any times it was taken off during the day. 

### 2.5. Data Processing

Data were downloaded using ActiLife version v6.13.4 (ActiGraph, Pensacola, FL, USA). We used ActiLife to generate an activity count file with 15-second epochs for each of the GT9X accelerometers in addition to the raw acceleration data. The files generated by ActiLife containing the 15 s epoch count data (.agd) were read into R using the R package PhysicalActivity 0.2–4 (https://CRAN.R-project.org/package=PhysicalActivity accessed on 1 May 2022); those containing raw acceleration data (.gt3x) were loaded using the R package GGIR version 2.9.0 (https://CRAN.R-project.org/package=GGIR accessed on 1 June 2022) [15]. GGIR was used to auto-calibrate the raw acceleration values [16] and calculate the average magnitude of dynamic acceleration corrected for gravity (the Euclidean norm minus 1 g); accelerations were averaged over the 15 s epochs. In-lab and at-home data were analyzed separately. Accelerometer data collected during the times indicated as wear time on the in-lab log sheet and take-home log sheet were considered wear time and included in the analyses. For each participant, in-lab counts recorded by the GT1M and GT9X were checked for lag between the devices. If lag was detected, the data were shifted to match the more accurate accelerometer; the more accurate device was the accelerometer that recorded activity counts of 0 at the times indicated to be rest periods by the in-lab log sheet. Lag was identified in 11 of the 20 in-lab sessions. Typically, the lag was 10 epochs or less; however, three sessions had a lag of 229 epochs between accelerometers.

We determined the time spent at the PA intensities for each participant’s at-home data using the Evenson cutpoints for count data [3] for 15 s epochs: SED < 25, LPA 26–573, MPA 574–1002, and VPA ≥ 1003. We calculated three IGs [4] for each participant using the at-home data collected by the GT9X (ActiGraph, Pensacola, FL, USA). The PA data were categorized into different bins on the basis of intensity and the log–log slope of the time at each intensity versus the intensity of each bin was calculated as the IG; all IGs were calculated using data from the GT9X. The first two IGs were calculated using the raw accelerations (accIG) and the 15 s epoch count (countIG) data, respectively. Data were sorted into 160 bins with a resolution of 25 mg for the raw data and 25 counts for the 15 s epoch data, with an additional bin including any counts of >4000 as originally described by Rowlands et al. [4]. When we calculated the countIG, more bins included data compared with the accIG; the ratio of bins including counts to bins including accelerations for ENMO ranged from 2.3–5.2. With more bins including data for the countIG, the intensity bin containing SED did not influence the countIG as much as the accIG. Thus, the countIG was generally shallower than the accIG. Therefore, we reduced the resolution of the bins for the count data to 100 counts per bin and created a third IG (adjusted IG; adjIG) using the 15 s epoch data with the number of bins reduced to 40. The adjIG included a similar number of bins with data points to the original accIG. One key difference was that for the countIG, the first bin included only sedentary activity (0–25 counts), while the first bin for the adjIG could include LPA as it also included count values from 26 to 100 [3]. Upon inspection of the first bin for the adjIG, 14% (SD = 6%) of the values were LPA, on average; therefore, its contents were mostly sedentary [4].

### 2.6. Statistical Analysis

We performed all analyses in RStudio (v2023.06.1+524; RStudio Team, 2020) with R (v.4.3.1) and considered *p*-values < 0.05 to be statistically significant. To compare correlations between the count data collected by the two accelerometers in the lab and at home, we calculated the coefficient of determination (R^2^) between the counts of the GT9X and GT1M. We used the R package rmcorr version 0.5.4 (https://CRAN.R-project.org/package=rmcorr accessed on 1 April 2022) to fit repeated-measures linear regressions [17], as the data had multiple measurements per participant. Bland–Altman (BA) plots assessed the agreement between the count data of the GT9X and GT1M for the in-lab data. We used the R package SimplyAgree version 0.1.2 (https://CRAN.R-project.org/package=SimplyAgree accessed on 1 June 2022) [18] for the repeated-measures BA analyses [19,20,21]. SimplyAgree could not handle the entire dataset for the at-home data; thus, we explored the agreement for three random participants on three random days. 

For the at-home data, we compared minutes per day spent at different PA intensities between the GT9X and GT1M. We calculated the coefficient of determination (R^2^) and used the intraclass correlation coefficient (ICC) and BA plots to assess the agreement between the GT9X and GT1M for minutes per day at each PA intensity. Intraclass correlation coefficient estimates and their 95% confidence intervals (CIs) were calculated using the R package psych version 2.4.3 (https://cran.r-project.org/web/packages/psych/index.html accessed on 1 April 2024) based on a single-measurement, absolute-agreement, two-way random effects model. We performed paired *t*-tests with equal variance to determine whether there was a difference in the number of minutes per day spent at different intensities of activity between the GT9X and GT1M’s home data. We assessed normality using Shapiro–Wilk tests and the equality of variance between samples using F-tests. 

Finally, to assess the agreement among the accIG, countIG, and adjIG, we used BA plots and calculated the 95% limits of agreement (LOA). Intraclass correlation coefficient estimates and their 95% CIs were calculated using the R package psych version 2.4.3 (https://cran.r-project.org/web/packages/psych/index.html accessed on 1 April 2024) based on a single-measurement, absolute-agreement, two-way mixed effects model.

## 3. Results

### 3.1. Descriptive Characteristics

Females (*n* = 15) and males (*n* = 5) aged 7–22 years old (M = 14.2, SD = 4.5), with a mean height of 156.3 cm (SD = 7.7) and mean body mass of 51.6 kg (SD = 18.8), participated in the study protocol. Participants were excluded if no valid data were collected from either accelerometer or if the GT9X accelerometer was incorrectly attached to its holder (i.e., the GT9X was not fully inserted into the belt clip), resulting in the vertical axis of the GT9X not being aligned with the vertical axis of the GT1M being worn at the same time. On the basis of these criteria, one participant was excluded from the in-lab analysis (an incorrectly attached GT9X) and three from the home analysis (one had an incorrectly attached GT9X, and two had no data from one or both of the accelerometers). We also excluded two participants from the in-lab analysis because they were given GT3X (ActiGraph, Pensacola, FL, USA) accelerometers to wear instead of GT1Ms. After exploratory analyses were conducted, an outlier in the at-home data was detected; the difference in the time spent in LPA (min/d) between the two accelerometers for this participant was ~100 min/d, as compared with a mean of 18 min/d for the whole group and a maximum of 47 min/d in the other participants. This participant was excluded from the at-home analyses, as the accelerometer stopped functioning (it would not connect to ActiLife) and was returned to the manufacturer. After these exclusions, 14 females and 3 males had complete in-lab data, and 11 females and 5 males had complete at-home data. The mean wear time for the at-home data was 732.2 min (SD = 123.3), with 510.4 min (SD = 126.4) and 505.9 min (SD = 128.9) in SED, 186.9 min (SD = 89.8) and 191.3 min (SD = 91.5) in LPA, 25.8 min (SD = 13.6) and 25.7 min (SD = 13.9) in MPA, and 9.0 min (SD = 8.0) and 9.3 min (SD = 7.9) for the GT1M and GT9X, respectively.

### 3.2. Comparisons of Count-Based Output for the Two Accelerometers

Bland–Altman (BA) plots are shown for the in-lab activities in Figure 2. There were strong correlations (R^2^ ≥ 0.70) between individual counts (15 s^−1^) measured by the two accelerometers for the combined in-lab data (Appendix A) and all individual activities performed during the in-lab protocol (Appendix A). The correlation between counts (15 s^−1^) measured by the two accelerometers across 3 days of at-home wear was moderate and weaker than the in-lab correlation (R^2^ = 0.50) (Appendix A). The BA plots showed moderate agreement between the two accelerometers for counts measured during all in-lab activities, with the largest mean difference being 24 counts for the obstacle course activity (Table 1). For the combined in-lab count data, agreement between the GT9X and GT1M was strong with little bias (bias = 2 counts) (Appendix A). BA plots for the individual in-lab activities did not show systematic or proportional bias for most activities; there was minimal proportional bias for jumping jacks and vertical jumps (Figure 2). Agreement for the combined at-home count data (bias = −2) (Appendix A) was similar to the agreement shown in the in-lab data.

The correlation between accelerometers was very strong for minutes per day spent at all PA intensities including LPA, MPA, VPA, total PA, and SED time (R^2^ > 0.95; Table 1). The mean time spent at each intensity (min/d) was not significantly different between the GT9X and GT1M (Table 2). The BA plots demonstrated moderate agreement between the accelerometers, with a mean difference of less than 5 min/d for each PA intensity (Table 2) and no visible pattern of systematic or proportional bias in the BA plots (Figure 3). 

### 3.3. The Intensity Gradient Using Counts Compared to Raw Accelerations

Intensity gradients were calculated for each participant: accIG (M = −2.347, SD = 0.231), countIG (M = −1.635, SD = 0.165), and adjIG (M = −2.192, SD = 0.187). The ICC of the accIG and the countIG (ICC (95% CI): 0.00 (−0.05, 0.12)) showed lower agreement than that between the accIG and adjIG (ICC (95% CI): 0.27 (0.14, 0.64)). The BA plots showed stronger agreement between the accIG and adjIG (bias = −0.154; 95% LOA [−0.648, 0.339]) than the accIG and countIG (bias = −0.712; 95% LOA [−1.293, −0.131]) (Figure 4). The BA plots showed more systematic bias between the accIG and countIG (with the countIG being consistently larger (shallower) than the accIG) than between the accIG and the adjIG (Figure 4).

## 4. Discussion

This validation study provided unique insight into whether the IG developed for ENMO data can be reproduced using count data. We first determined that counts (15 s^−1^) and minutes per day at different intensities of PA were consistent between the two accelerometers. Although individual counts were moderately correlated in the free-living at-home data, minutes per day at various intensities were strongly correlated and comparable between models. We also investigated whether the IG, a PA metric designed for raw acceleration data, could be used with count data. We found that with the GT9X’s data, the adjIGs generated from the count data were similar to the accIGs derived from the ENMO data.

Our results indicated that the count data collected by the GT9X and GT1M accelerometers were more strongly correlated when measuring consistent, repetitive PA. For in-lab activities, the counts were most strongly correlated during the consistent activities of brisk walking and jogging. Furthermore, we found that the combined in-lab count data were more strongly correlated than the at-home count data. During in-lab testing, the participants were performing activities at a consistent intensity for several minutes at a time and were active for at least half the time the accelerometer was worn. In contrast, during free-living wear, the intensity of activity varied substantially, and more accelerometer wear time was spent in SED and LPA. The stronger correlation seen between the GT9X and GT1M for the combined in-lab data compared with the combined at-home data is likely to be due to the consistent activity at higher intensities that took place during the in-lab sessions. Our findings are consistent with those of multiple other studies that found that counts (min^−1^) did not differ between four generations (7164 and three versions of GT1M) of ActiGraph accelerometers [10], nor between the GT1M and GT3X accelerometers on the vertical axis [12] during walking and running. Additionally, we observed a similar correlation for counts measured at home between accelerometers, as did a study examining the average correlation between the counts/min/day of the ActiGraph 7164 and wGT3X-BT for free-living activity (r = 0.74, *n* = 87) [22]. 

We found that the free-living PA intensity data (min/d) were comparable between the GT9X and GT1M accelerometers. Although no previous studies compared the GT9X and GT1M ActiGraph accelerometers, other studies identified good agreement between the GT1M and GT3X ActiGraph models [11] and between the 7164 and GT1M ActiGraph models [9] when comparing time spent at different PA intensities in minutes per day. Further, the differences we observed between the GT9X and GT1M were similar to the differences observed between two GT3X+ accelerometers worn on contralateral hips using 10 s epochs (LOA: ±18.2 min, ±6.3 min, ±3.5 min, and ±51 min for LPA, MPA, VPA, and SED, respectively). This indicates that the differences we found between the GT9X and GT1M were similar to those found between two accelerometers of the same model [23]. However, some studies have found disagreement between accelerometer models, even for time spent at different PA intensities. Differences in minutes per day of SED and LPA were found between the 7164 and GT1M using 60 s epochs (~20 min/d less in SED and more in LPA), [24] and differences in minutes per day in LPA, MPA, MVPA, and SED were found between the 7164 and wGT3X-BT using 60 s epochs (mean differences ranging from −3.9 to 12.1 min/d) [22]. All studies identified good agreement between accelerometers for minutes per day spent in VPA. 

We were also able to reproduce the IG metric created for use with raw acceleration data with count-based data. Following adjustment of the bins’ sizes for the adjIG, the mean bias decreased and the 95% limits of agreement narrowed. Thus, it is important to consider the appropriate bin size when applying the IG to different accelerometer metrics. This newly developed IG for use with count data may be valuable to researchers with historical accelerometry datasets who are interested in applying cutpoint-free metrics to their data. The IG has several strengths compared with traditional cutpoint metrics. First, cutpoint metrics do not quantify free-living PA across the whole spectrum of PA intensity, whereas the IG does. Therefore, using the IG could allow researchers to better identify how an individual’s spread of PA across the intensity spectrum is associated with different health outcomes. Second, with cutpoint metrics, there is no differentiation between the differing magnitudes of acceleration contained within one PA intensity bin. For example, a participant who spent most of their time at the high end of MVPA would have the same minutes per day in MVPA as someone who spent all their time at the very low end of the MVPA intensity bin defined using cutpoints. However, their IGs would differ, with the participant spending more time at the high end of MVPA having a shallower slope than the one who spent most of their time at the low end. Therefore, validating that the IG can be reproduced with count data provides researchers with count-based datasets an opportunity to glean more information from data that have been analyzed using cutpoint methods in the past.

A limitation of this study is that the idle sleep mode was turned on for some of the GT9X accelerometers to preserve the battery’s life, as the battery did not last long enough for at-home data collection. Enabling the idle sleep mode has been found to affect the minimum, maximum, and range of values output by the accelerometer as compared with when it is turned off [25]. We also used 15 s epochs rather than the 5 s epochs that were used when the IG was initially designed [4]. This may have contributed to the lower IG seen with the count IG, as previous work has shown that as the epochs’ size increases, the IG becomes shallower [26]. Furthermore, the lowest intensity bins in the adjIG likely differed in the nature and range of activity compared with the accIG, as the adjIG bins included more values above the sedentary threshold than the first bins in the accIG. Additionally, due to the volume of data and limitations in the R packages used to analyze the raw accelerometer data, we were not able to use the complete at-home dataset to determine the agreement between the two accelerometers for the at-home data. We did not determine an a priori criterion for clinically relevant limits of agreement to define comparability between metrics. Finally, the small sample size may limit the generalizability. Several strengths of this study should also be noted. Both in-lab and at-home data were collected, allowing us to compare the data collected by the accelerometers in both the structured lab and free-living settings. Furthermore, this study is the first, to our knowledge, to investigate whether a PA metric designed for raw accelerations can be used with count data. 

As accelerometers continue to be developed and upgraded, there is a need to understand whether the accelerations they measure are comparable so that we know whether studies conducted using different models of accelerometer (e.g., the GT1M versus GT9X) can use the same metrics to quantify PA data. Specifically, it would be valuable to know whether the novel metrics created to quantify PA data developed for use with raw acceleration data can be used with the count data that historical datasets contain. By examining accelerometers’ output in structured the in-lab and unstructured free-living settings, we determined that the counts (15 s^−1^) were comparable between the GT1M and GT9X for structured but not unstructured PA. However, traditional cutpoint metrics (min/d at various intensities) were comparable with count-based free-living PA. Furthermore, we determined that the IG, originally developed for use with raw acceleration data, was reproducible with count data. Collectively, our findings suggest that free-living count-based PA data are comparable between the GT9X and GT1M when quantified using both traditional cutpoint metrics and the newly introduced IG metric. In particular, the International Children’s Accelerometry Database (ICAD) includes 44,454 count-based PA accelerometry files collected from children and adolescents (aged 3–18 years) in 20 studies worldwide that could now be assessed using the intensity gradient [27]. Applying this new IG metric may provide additional insight into PA and its associations with health outcomes using these existing databases, without the need to collect new raw acceleration data. Future work should test the feasibility of using other metrics previously designed for raw accelerations with count data. 

## Figures and Tables

**Figure 1 sensors-24-03019-f001:**
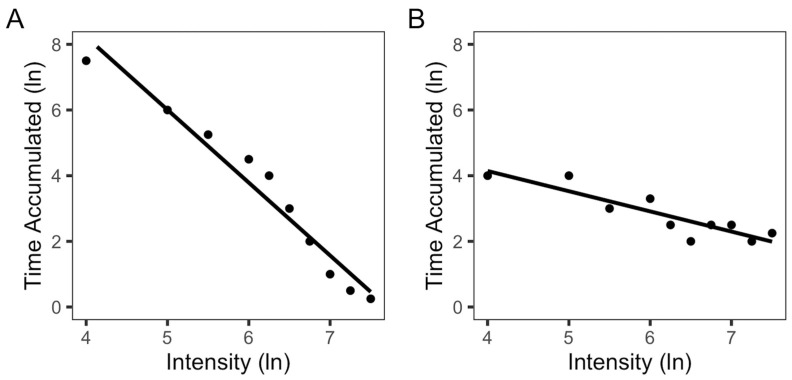
Two examples of the intensity gradient. Time accumulated in each bin is represented as the black dots. (**A**) Someone with more time at lower intensities of physical activity has a steeper slope. (**B**) A person with more time across the full spectrum of physical activity has a shallower slope.

**Figure 2 sensors-24-03019-f002:**
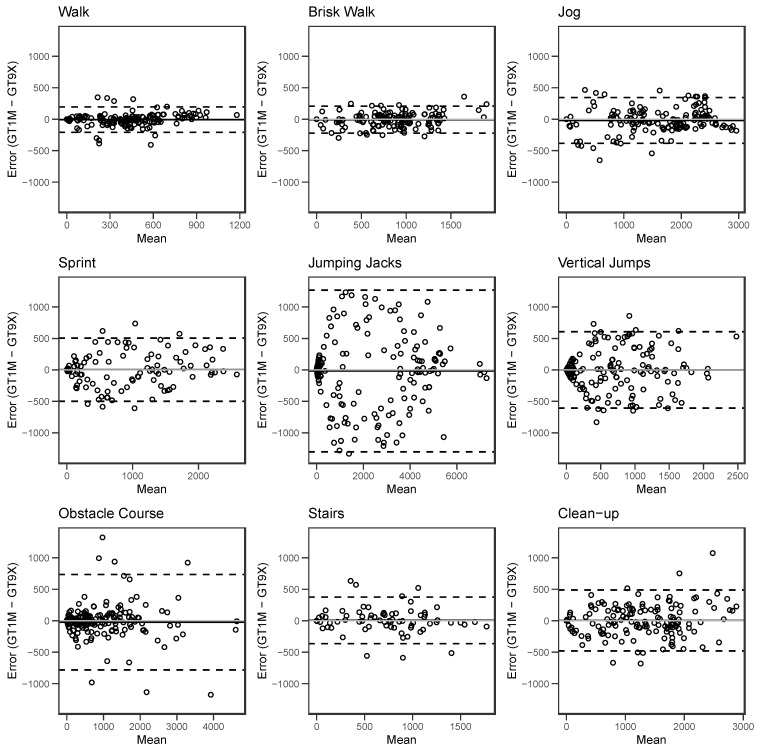
Bland–Altman plots for each individual activity performed during the in-lab testing session. Data from all participants were combined. Plots show the individual participant data points as the dots, mean difference as the solid black line, the zero line as the solid grey line, and the upper and lower 95% limits of agreement as dotted lines. The mean difference and limits of agreement were calculated using repeated-measures Bland–Altman analysis [19,20,21].

**Figure 3 sensors-24-03019-f003:**
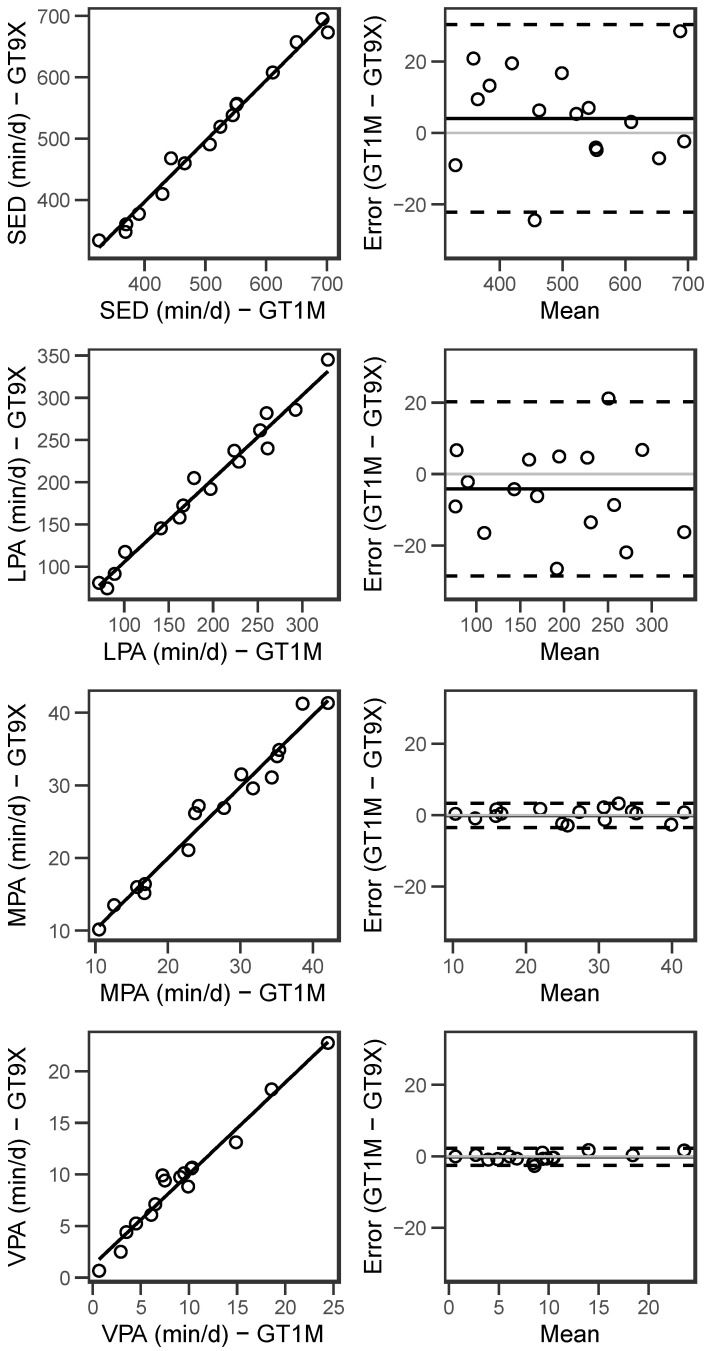
Linear regression and Bland–Altman plots for minutes per day at each physical activity intensity for at-home accelerometer data. Data from all participants were combined and individual participant data points are represented as dots. Linear regressions show the line of best fit; the Bland–Altman plots show the mean difference as the solid black line, the zero line as the solid grey line, and the upper and lower 95% limits of agreement as dotted lines.

**Figure 4 sensors-24-03019-f004:**
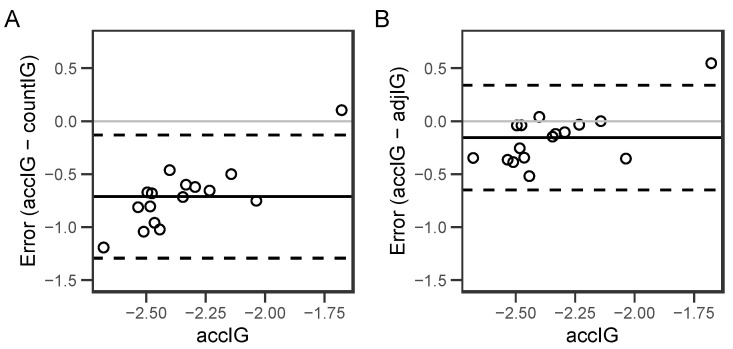
Bland–Altman plots comparing the agreement between the accIG and the two IGs calculated using count data. (**A**) Bland–Altman plot for the accIG and countIG (the intensity gradient calculated using 160 bins and the count data) (**B**) Bland–Altman plot for the accIG and adjIG (the intensity gradient calculated using 40 bins and the count data). Individual participant data points are represented as the dots, the mean difference is shown as the solid black line, the zero line as the solid grey line, and upper and lower 95% limits of agreement as the dotted lines.

**Table 1 sensors-24-03019-t001:** Linear regression and Bland–Altman analysis comparing the correlations and agreement between activity counts measured by the GT9X and GT1M in the lab.

	Linear Regression			Bland–Altman		
	Slope	Intercept	R^2^	Mean Difference (Counts)	Upper LOA	Lower LOA
Combined	0.94	33.30	0.92	2	552	−549
Walk	0.82	78.57	0.75	−5	196	−207
Brisk walk	0.90	81.06	0.92	−6	208	−220
Jog	0.90	162.55	0.94	−19	344	−382
Sprint	0.90	81.09	0.84	2	505	−500
Jumping jacks	0.91	193.74	0.88	−17	1266	−1300
Vertical jumps	0.81	118.01	0.70	−1	606	−608
Obstacle course	0.85	157.92	0.78	−24	734	−783
Stairs	0.92	45.76	0.81	5	375	−365
Cleaning up	0.84	198.17	0.83	3	488	−482

Abbreviation: LOA, limit of agreement. N = 17.

**Table 2 sensors-24-03019-t002:** Linear regression, Bland–Altman analysis, and *t*-tests, comparing the correlations and agreement between time at each activity intensity based on measurements by the GT9X and GT1M at home.

	Linear Regression			Bland–Altman			ICC
	Slope	Intercept	R^2^	Mean Difference (min/d)	Upper LOA	Lower LOA	ICC (95% CI)
Combined (counts)	0.71	28	0.50	−2 *	−429 *	426 *	
LPA (min/d)	0.99	6.57	0.98	−4.13	20.25	−28.50	0.99 (0.97, 0.99)
MPA (min/d)	0.98	0.33	0.97	−0.11	3.28	−3.50	0.98 (0.96, 0.99)
VPA (min/d)	0.89	1.17	0.96	−0.11	2.28	−2.50	0.97 (0.95, 0.99)
Total PA (min/d)	0.99	7.19	0.98	−4.08	22.16	−30.32	0.99 (0.98, 0.99)
SED (min/d)	0.99	0.95	0.99	4.08	30.32	−22.16	0.99 (0.98, 1.00)

Abbreviations: LOA, limit of agreement; CI, confidence interval; LPA, light physical activity; MPA, moderate physical activity; VPA, vigorous physical activity; total PA, total physical activity (LPA, MPA, and VPA); SED, sedentary. N = 16. * Bland–Altman analysis for three random participants on three random days.

## Data Availability

The raw data supporting the conclusions of this article will be made available by the authors upon reasonable request.

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
