# Peer review of "Adapting the Intensity Gradient for Use with Count-Based Accelerometry Data in Children and Adolescents"

_sensors, 2024, doi:10.3390/s24103019_

Round 1

Reviewer 1 Report

Comments and Suggestions for Authors

See attached file

Reviewer 2 Report

Comments and Suggestions for Authors

The structure and writing style of this article is different from the average article, the article is narrating the conduct of the experiment and the results of the experiment throughout the article without any description of the methodology, it is not clear how it differs from the previous research and where the contribution lies. The paper lacks some relevant background description and introduction, as well as a comprehensive analysis of the current state of research.

Here are some specific questions:

1. Please provide more background and current status of research on gradient intensity.

2. Maybe you can provide an overall system framework or figures to enrich your contents.

3. "This metric was developed for use with raw accelerations (gravitational units) and has not been validated for use with count-based accelerometer data. In this study, we determined whether the intensity gradient could be reproduced using count-based accelerometer data. " This sentence is unconvincing.

  Comments on the Quality of English Language

Need to improve

Reviewer 3 Report

Comments and Suggestions for Authors

The topic is very inteersting and important for applications in medicine, sport sciences and psychology. The presentation is clear. The literature review includes the most appropriate publications. 

The only remark: the quality of Fig 2 must be improved to the size equivalent to the sketches in Figures 2-4. 

Author Response

 We thank the Reviewer for their suggestions.

  1. Comment: The topic is very interesting and important for applications in medicine, sport sciences and psychology. The presentation is clear. The literature review includes the most appropriate publications.

Response: Thank you for your comment we appreciate the feedback.

  1. Comment: The quality of Fig 2 must be improved to the size equivalent to the sketches in Figures 2-4.

Response: Thank you for your comment, we have adjusted the size of the sketches in Figure 2 by removing the linear regressions to supplementary materials.

Page 6 – Updated figure 2

Page 11, Lines 498-500. Figure S2: Linear regression plots for each individual activity performed during the in-lab testing session. Data from all participants are combined. Plots show the line of best fit based on multiple linear regression [12].

Round 2

Reviewer 1 Report

Comments and Suggestions for Authors

Thank you for careful attention to the comments and revisions to the manuscript. It is much clearer and I only have a few further minor suggestions.

Lines 181-182. I appreciate the further analyses and that only 14% of the first bin is LPA but suggest a bit more caution in wording as the intensity range of these bins is quite different between the adjIG and accIG.

I suggest delete:and similar to that of the first bin used for the accIG’

and add in the limitations something along the lines of ‘the lowest intensity bins in the adjIG likely differed in the nature and range of activity captured. i.e., the first bin for the adjIG was double the size of commonly used inactive thresholds (50 cpm). In contrast the first bin for the accIG was approx. 2/3 the size of commonly used inactive thresholds (40 mg), with 3 bins (75 mg) still not reaching double the inactive threshold.

Line 249 / Table 2: Please specify type of ICC and in table 2 present 95% CI for ICC, see Koo and Li 2016, https://pubmed.ncbi.nlm.nih.gov/27330520/

Section 3.4 Please present ICC for IG comparison

Abstract line 27. Given there was no a priori consideration of what was ‘comparable’, it is difficult to justify calling them ‘comparable’. Consider sticking with stating what you have done. E.g. ‘We have generated bin sizes that can be used to generate IG from count data. The mean bias (95% limits) is ….’

Line 352 – similar to above, ‘comparable’ isn’t really appropriate. Perhaps could state something along the lines of:

following adjustment of bin sizes for the adj IG the mean bias decreased and 95% limits narrowed demonstrating that it is important to consider the appropriate bin size when applying the IG to different accelerometer metrics

Limitations: Consider adding as a limitation that 15 s was used as the epoch rather than 5s. As bin size increases IG gets higher and the ability to detect associations with health decreases, see  https://www.ncbi.nlm.nih.gov/pmc/articles/PMC9982530/. Thus, epoch size may have contributed to the lower IG you saw with the count IGs.

Reviewer 2 Report

Comments and Suggestions for Authors

The revised manuscript has been refined and modified to fix some previous problems. I do suggest putting some visual illustrations of the system or experimental procedures to increase readability and interest.

Author Response

Unfortunately, we do not understand the Reviewers's request as we already include a visual of the primary concept of the manuscript (intensity gradient) as well as figures throughout.